# GTN Enhances Antitumor Effects of Doxorubicin in TNBC by Targeting the Immunosuppressive Activity of PMN-MDSC

**DOI:** 10.3390/cancers15123129

**Published:** 2023-06-09

**Authors:** Nesrine Mabrouk, Cindy Racoeur, Jingxuan Shan, Aurélie Massot, Silvia Ghione, Malorie Privat, Lucile Dondaine, Elise Ballot, Caroline Truntzer, Romain Boidot, François Hermetet, Valentin Derangère, Mélanie Bruchard, Frédérique Végran, Lotfi Chouchane, François Ghiringhelli, Ali Bettaieb, Catherine Paul

**Affiliations:** 1Laboratoire d’Immunologie et Immunothérapie des Cancers, EPHE, PSL Research University, 75006 Paris, France; nesrine.mabrouk@inserm.fr (N.M.); cindy.racoeur@u-bourgogne.fr (C.R.); aurelie.massot4@gmail.com (A.M.); silvi.ghio50@gmail.com (S.G.); malorie.privat@u-bourgogne.fr (M.P.); lucile.dondaine@u-bourgogne.fr (L.D.); ali.bettaieb@u-bourgogne.fr (A.B.); 2LIIC, EA7269, Université de Bourgogne Franche Comté, 21000 Dijon, France; 3Genetic Intelligence Laboratory, Weill Cornell Medicine-Qatar, Qatar Foundation, Doha P.O. Box 24144, Qatar; jis2015@qatar-med.cornell.edu (J.S.); loc2008@qatar-med.cornell.edu (L.C.); 4Plateforme de Transfert en Biologie Cancérologique, Centre GFL Leclerc, 21000 Dijon, France; eballot@cgfl.fr (E.B.); ctruntzer@cgfl.fr (C.T.); vderangere@cgfl.fr (V.D.); frederique.vegran@inserm.fr (F.V.); fghiringhelli@cgfl.fr (F.G.); 5Unit of Molecular Biology, Georges-François Leclerc Cancer Center—UNICANCER, CNRS UMR 6302, 21000 Dijon, France; rboidot@cgfl.fr; 6CRI UMR INSERM1231, 21000 Dijon, France; francois.hermetet@u-bourgogne.fr (F.H.); melanie.bruchard@gmail.com (M.B.); 7UBFC, 21000 Dijon, France

**Keywords:** nitric oxide donor, glyceryl trinitrate, doxorubicin, immunotherapy, triple-negative breast cancer, PMN-MDSC

## Abstract

**Simple Summary:**

The immune system plays an important role in the effectiveness of chemotherapy used in the treatment of different types of breast cancer (BC). However, this system is often deficient in most cancers, including BC. Thus, any strategy aimed to reactivate the immune system should amplify the effectiveness of treatments. The aim of our study is to test the beneficial effect of glyceryltrinitrate (GTN) a nitric oxide donor (used in the treatment of angina pectoris) when associated with doxorubicin, a chemotherapy used in the treatment of BC, especially the triple-negative type (TNBC). This work was performed using a mouse model of TNBC. We have shown that the combination of the two drugs induces greater tumor regression than the monotherapies, an effect linked to a stronger activation of the immune system correlated with a decrease in immunosuppressive immune cells. These findings suggest a therapeutic perspective in the treatment of TNBC.

**Abstract:**

(1) Background: Immunosuppression is a key barrier to effective anti-cancer therapies, particularly in triple-negative breast cancer (TNBC), an aggressive and difficult to treat form of breast cancer. We investigated here whether the combination of doxorubicin, a standard chemotherapy in TNBC with glyceryltrinitrate (GTN), a nitric oxide (NO) donor, could overcome chemotherapy resistance and highlight the mechanisms involved in a mouse model of TNBC. (2) Methods: Balb/C-bearing subcutaneous 4T1 (TNBC) tumors were treated with doxorubicin (8 mg/Kg) and GTN (5 mg/kg) and monitored for tumor growth and tumor-infiltrating immune cells. The effect of treatments on MDSCs reprogramming was investigated ex vivo and in vitro. (3) Results: GTN improved the anti-tumor efficacy of doxorubicin in TNBC tumors. This combination increases the intra-tumor recruitment and activation of CD8^+^ lymphocytes and dampens the immunosuppressive function of PMN-MDSCs PD-L1^low^. Mechanistically, in PMN-MDSC, the doxorubicin/GTN combination reduced STAT5 phosphorylation, while GTN +/− doxorubicin induced a ROS-dependent cleavage of STAT5 associated with a decrease in FATP2. (4) Conclusion: We have identified a new combination enhancing the immune-mediated anticancer therapy in a TNBC mouse model through the reprograming of PMN-MDSCs towards a less immunosuppressive phenotype. These findings prompt the testing of GTN combined with chemotherapies as an adjuvant in TNBC patients experiencing treatment failure.

## 1. Introduction

The triple-negative breast cancer (TNBC) subtype, representing 11.2% of new breast cancer (BC) cases, is considered the most refractory BC and is responsible for the majority of the disease-related deaths [1]. To date, chemotherapy remains the standard of care since therapies directed against specific molecular targets have not significantly improved survival in patients with TNBC [2]. While these treatments are very effective in TNBC patients diagnosed at an early stage, in those who develop metastatic disease, the median of overall survival (OS) remains low (13–18 months) [3]. In recent years, the many advances realized in the understanding of the mechanisms of tumor immunology, as well as the development of immunotherapy, have offered new perspectives in the management of TNBC. Indeed, an increasing body of data suggest that the outcome of TNBC is dependent on the immune system. The rate of tumor-infiltrating lymphocytes (TILs) has been shown to be predictive of the response to treatment with neoadjuvant or adjuvant chemotherapies and is strongly associated with an improvement in patient survival [4,5]. More recently, the TNBC subtype has also been described as exhibiting higher expression of Programmed Death-Ligand 1 (PD-L1), an immune checkpoint molecule which contributes to immune evasion [6]. Although the clinical evaluation of inhibitors of these checkpoints has shown modest efficacy in monotherapy, their combination with chemotherapies provides encouraging results [2,7]. Indeed, the results of a large phase III trial (IMpassion130) showed that in the PD-L1 positive TNBC population, the median of OS was increased in the cohort receiving nab-paclitaxel in combination with atezolizumab (a PD-L1 inhibitor, 12.9 months) compared to the cohort receiving chemotherapy only (3.7 months) [8,9]. Another field of investigation is based on the combination of chemotherapies. Therefore, clinicians often use sequential anthracycline and taxane regimens [3]. Indeed, increasing the effectiveness of these chemotherapies remains a major issue in the management of patients with TNBC. To improve the anti-tumor effect of chemotherapies, a combination with a nitric oxide (NO) donor such as the glyceryl trinitrate (GTN) is a possibility. GTN is a drug widely used in the treatment of certain heart disease as angina pectoris [10]. Indeed, several clinical trials have demonstrated the beneficial effects of the use of GTN, in combination with standard chemotherapies associated or not with radiotherapy, in patients with a non-small cell lung cancer or prostate cancer [11,12,13]. Studies from our laboratory have also described the anti-tumor activity of NO by showing the pro-apoptotic role of GTN in human colon cancer cells [14,15]. Apoptosis sensitization of colon and mammary cancer cells to ligand of Fas (FasL) by GTN, via Fas S-nitrosylation, has also been demonstrated [16]. It should be noted that adverse effects of GTN combined with chemotherapy in cancer patients are minimal; these are reflected in more headaches, rare cases of moderate hypotension (grade 1) but no severe hypotension, and no other side effects have been reported [11].

In this study, we explored the combination of GTN with doxorubicin in mouse models of BC. We demonstrated that GTN synergizes with doxorubicin to inhibit tumor growth. Such an effect was correlated with a GTN-mediated change in tumor immune cell infiltration leading to an anti-tumor microenvironment and an effective immune response. In particular, we demonstrated that the GTN/doxorubicin combination reprograms polymorphonuclear (PMN)-MDSCs to a less immunosuppressive phenotype within the tumor microenvironment.

## 2. Materials and Methods

### 2.1. Tumor Models and Treatments

Balb/C female mice of 6–8 weeks of age were purchased from Charles River Laboratories. All experiments followed the guidelines of the Federation of European Animal Science Associations and were approved by the Ethics Committee of Burgundy University (Dijon, France) and by the Ministry of Agriculture and Food of France (Apafis #10666). Mice previously anesthetized with isoflurane were subcutaneously injected with 5 × 10^5^ mammary cancer cells (4T1 or EMT6). Subcutaneous treatment with GTN (Nitronal, 5 mg/kg in 100 µL of NaCl) [17] or physiological serum (NaCl, Ctrl) started one day after tumor cells injection and was renewed 3 times a week. Intravenous injection of doxorubicin (8 mg/kg in 100 µL of NaCl) or NaCl started once tumors reach an average volume of 50 mm3 (~day 8) and was repeated once every 7 days. Intraperitoneal injection of anti-PD-1 antibody (clone RMP1-14, Bioxcell (Euromedex, Mundolsheim, France), 10 mg/kg in 100 µL of NaCl) or NaCl started the day after the first injection of doxorubicin and was renewed twice a week. For CD8^+^ T lymphocytes depletion, intraperitoneal injection of anti-CD8a (Clone 2.43, Bioxcell, 5 mg/mL, diluted in 100 µL of NaCl) or IgG2a isotype control (BioxCell, 5 mg/mL in 100 µL of NaCl) started the day before the first injection of doxorubicin and was renewed once a week. Treatment with an antioxidant, the N-acetyl-L-cysteine (NAC, 3.5 mg/mL, Sigma Aldrich, Saint-Quentin-Fallavier, France), diluted in the drinking water, started the day after tumor cell injection and was renewed 3 times a week, as previously described [18]. Tumor growth monitoring was performed every 2–3 days using a caliper. The tumor volume (mm^3^) is determined with the formula (l × l × L)/2. Animals were sacrificed once the tumor volume reached 2000 mm^3^.

### 2.2. Cell Lines and Treatments

Mouse breast cancer cells 4T1 (TN) and EMT-6, all syngeneic from Balb/C mice, were obtained from the American Type Culture Collection (ATCC) and were tested regularly for mycoplasma contamination using a Universal Mycoplasma detection kit (ATCC). These cells were maintained in DMEM medium (Dominique Dutscher, Bernolsheim, France) supplemented with 10% fetal bovine serum (FBS, Dominique Dutscher) at 37 °C in a humid atmosphere with 5% CO_2_. MSC2, a murine immortalized MDSC cell line obtained from Balb/C Gr-1+ splenocytes, was obtained from V. Bronte (Istituto Oncologico, Padova, Italy). They were cultured in RPMI 1640 (Dominique Dutscher) supplemented with 10% FBS and 1% penicillin, streptomycin and amphotericin B (PSA) antibiotic mix (PAN Biotech GmbH, Aidenbach, Germany) at 37 °C under 5% CO_2_.

For ex vivo experiments, MDSCs and/or anti-CD3/anti-CD28 Dynabeads (Gibco, Thermo Fischer Scientific, Waltham, MA, USA)-activated T lymphocytes were treated or not (Ctrl) for 24, 48 h or 5 days with 100, 250 or 500 nM of doxorubicin +/− 10 or 100 µM of GTN (Merck, Lyon, France) +/− 10 mM of NAC. MSC-2 cells were also treated with 1 mM of Nitrosocysteine (Cys-NO) for 15 min.

### 2.3. Isolation of Primary Mouse Immune Cells

PMN-MDSCs were isolated from Balb/C mice injected with 4T1 breast tumor cells and treated with NaCl (Ctrl) +/− doxorubicin +/− GTN. At day 23, the tumors were recovered and PMN-MDSCs were isolated as previously described [17]. PMN-MDSCs were cultured alone or co-cultured for 5 days with T lymphocytes activated by anti-CD3/anti-CD28 beads and labeled with cell traceTM violet (Invitrogen, Waltham, MA, USA). T lymphocytes were previously isolated from spleens and lymph nodes of naive mice using the Pan T cells isolation kit II (Miltenyi, Bergisch Gladbach, Germany).

### 2.4. RT-qPCR

Mice tumors were collected and lysed in Trizol reagent (Ambion—Thermo Fischer Scientific) and total RNA was extracted. one hundred to five hundred nanograms nanograms of RNA were reverse-transcribed into cDNA using M-MLV reverse transcriptase, random primers and recombinant RNasin^®^ Plus RNase Inhibitor (Promega, Charbonnières-les-Bains, France). cDNA was quantified via real-time PCR using PowerUpTM SYBRTM Green (Applied Biosystems, Thermo Fischer Scientific) on Applied Biosystems ViiA™ 7 Real-Time PCR System (Thermo Fischer ScientificRelative mRNA levels were determined using the 2-ΔCt method after actin or PPiA subtracting (ΔCt). Forward and reverse primers used (Eurogentec, Seraing, Belgium) are listed in the Appendix A.

### 2.5. Flow Cytometry Analysis

To determine the CD8 TILs and MDSCs tumor infiltration, the tumors were dissociated and red blood cells removed [19]. One million cells were stained with antibodies (Appendix A) before flow cytometry analysis.

For the in vitro and ex vivo analysis, 24 or 48 h after in vitro treatments, cells were recovered and centrifuged, then pellets were taken up in phosphate-buffered saline for the following analyses.

Annexin V-AAD labelling was performed according to the manufacturer’s protocol (BD Biosciences, Le Pont de Claix, France). For the cell membrane FATP2 determination, cell pellets were taken up in 50 μL of Flow Cytometry Staining Buffer (FCSB, eBioscience) for 10 min. Fixable Viability Stain 700 (FVS700, BD Bioscience, 1:10,000) and FATP2 primary antibody (Abcam, 1:50) staining was performed before incubation with 0.25 % paraformaldehyde (Electron Microscopy Science) and labelling with the Donkey anti-rabbit IgG H&L secondary antibody (Alexa Fluor 488, Abcam 1:500). Cells were then analyzed by flow cytometry. For the reactive oxygen species (ROS) detection, pellets were incubated or not with carboxy-2′,7′-dichloro-dihydro-fluorescein diacetate (DCFH2DA, 1 mM) probe according to the manufacturer’s protocol (Sigma). For the PD-1 and PD-L1 labelling, pellets were taken up in 50 μL of FCSB for 10 min before staining with FVS700 (BD Bioscience) and, respectively PD-1 or IgG2b APC (eBioscience) and PD-L1 or IgG2b PE (Biolegend).

All acquisitions were performed on a BD Canto cytometer equipped with BD FACSDiva software (BD Biosciences), and data analyzed with FlowJo software 10.8.1.

### 2.6. Immunohistochemistry

Tumors collected from Balb/C mice bearing 4T1 tumors were cut in 4 µm thick slices after formalin fixation and paraffin embedding. After antigen retrieval, the slices were labeled for 1 h with the rabbit anti-CD8 polyclonal antibody (abcam, 1:100), then stained with goat anti-rabbit Alexa Fluor 647secondary antibody (Invitrogen, Thermo Fischer Scientific). Nuclei were stained with a DAPI solution (Spectral DAPI, PerkinElmer, Villebon-sur-Yvette, France) and slices mounted in Prolong Diamond Antifade (Thermo Fisher) before imaging and analysis. For each slide, five to ten representative areas were imaged using a Mantra Quantitative Pathology Workstation (PerkinElmer). After nuclear recognition and phenotype learning, phenotyping of CD8^+^ T cells were achieved using the inForm Cell Analysis software (PerkinElmer).

### 2.7. RNAseq

The quantity and quality of mouse Trizol RNA extracted from Balb/C mice bearing 4T1 tumors and treated or not with doxorubicin +/− GTN were measured using Agilent 2100 Bioanalyzer (Agilent Technologies Inc., Santa Clara, CA, USA). Four thousand nanograms of RNA from each sample were used to prepare cDNA library using the TruSeq™ sample preparation kit v2 (Illumina, Inc., San Diego, CA, USA). Briefly, poly-A selected mRNAs were first conversed to single-stranded cDNA using random hexamer primer and followed by second-strand generation. Sequencing adapters were then ligated to the fragmented cDNA, followed by PCR amplification. Six cDNA libraries were pooled per lane of the flow cell for 100 base pair single-end sequencing on a HiSeq 4000 sequencer (Illumina, Inc., San Diego, CA, USA). The raw sequencing data were stored in FASTQ format. The paired-end reads from FASTQ file were aligned using STAR aligner version 2.6.0a [20] to Genome Reference Consortium Mus musculus genome reference build 38 (GRCm38) as the reference genome. We used RSEM version 1.3.0 [21] to estimate transcripts per million (TPM) and raw read counts values using Musmusculus Ensembl gene annotation as downloaded from iGenome’s (Illumina^®,^ Evry, France).

### 2.8. Biotin Switch Assay (BSA)

MSC2 cells (16. 10^6^) were treated with GTN +/− doxorubicin or Cys-NO for 24 h. BSA was performed as previously described [16]. The supernatants were collected to be analyzed for STAT5-S-nitrosylation by 10 % SDS-PAGE and Western blot.

### 2.9. Western Blot

Ice-cold RIPA buffer (15 mM NaCl, 50 mM Tris HCL PH 7.4, 1 mM EDTA, 0.5 mM EGTA, 1% Triton, 0.1% SDS, 0.1% sodium deoxycholate) in the presence of proteases (Roche) and phosphatases (Sigma) inhibitors cocktails was used to cell lysis. After 30 min of ice incubation, the lysates were cleared via centrifugation at 16,000 g for 15 min at 4 °C, and the protein concentration was determined using the BioRad Dc protein assay as recommended by the manufacturer (Bio-Rad, Marnes-La-Coquette, France). Proteins were mixed with 5X Laemmli loading buffer and incubated 10 min at 95 °C. Fifty μg of proteins were migrated in a 10% SDS-PAGE and transferred in a nitrocellulose membrane. The membranes were probed overnight at 4 °C with primary specific antibodies (STAT5, pSTAT5 (Cell Signaling, Danvers, MA, USA), HSC70 (Santa Cruz, Santa Cruz, CA, USA) and FATP2 (Abcam, Cambridge, UK)) after 45 min of non-specific binding sites blocking with TBS 0.1% Tween 5% nonfat milk (pSTAT5) or bovine serum albumin (STAT5, FATP2 and HSC70). The membranes were washed and incubated for 1 h at room temperature with horseradish peroxidase-conjugated secondary antibody (Jackson Immunoresearch Laboratories). Immunoblot was revealed using Clarity Western ECL Substrate kit and ChemiDoc imaging system (Bio-Rad).

### 2.10. Statistical Analysis

The results are shown as means ± SEM, and statistical analysis was performed using paired (in vitro and ex vivo experiments) unpaired (in vivo) one- or two-tailed Student’s *t*-test with significance determined at *p* ≤ 0.05. Tumor growth was evaluated using two-way ANOVA with Bonferroni correction for multiple comparisons. RNAseq differential expression analysis was performed with the DESeq2 R package [22]. Four groups were compared to each other. Raw *p*-values associated with each gene were adjusted using Bonferroni correction, as advised. Differential expression analyses were performed using the R software (http://www.R-project.org/ accessed on 1 October 2020). GraphPad Prism 7 was used to perform all statistical calculations. Differences are considered statistically significant at * *p* ≤ 0.05; ** *p* ≤ 0.01; *** *p* ≤ 0.001, and **** *p* ≤ 0.0001.

## 3. Results

### 3.1. Addition of a NO Donor (GTN) Strongly Increases Doxorubicin Anti-Tumor Efficacy

In order to test whether the NO donor GTN can improve the therapeutic efficacy of a chemotherapy conventionally used in the treatment of BC, we used the doxorubicin/GTN combination in two BC models (Figure 1A–C), the triple-negative 4T1 model and the estrogen and progestin receptor positive cell EMT-6 model. The latter model is a less aggressive tumor model than the triple-negative model. We showed that doxorubicin slows triple-negative 4T1 tumor growth (Figure 1B) less efficiently than EMT-6 tumors, an estrogen and progestin receptor positive cell line [23] (Figure 1C), while GTN alone shows no effect. However, adding GTN to a doxorubicin treatment significantly improved its anti-tumor efficacy, almost completely inhibiting tumor progression in the EMT-6 BC model (Figure 1C). As previous work showed that GTN alone or in combination with FasL induced a high toxicity of human colon cancer cells in vitro, the effect of this combination of treatments on BC cells has been investigated. While doxorubicin alone, but not GTN, induced TNBC cell death, no further cytotoxicity was demonstrated with the doxorubicin/GTN combination (Figure 1D). However, drugs, when used alone or in combination, failed to induce cell death in the in vivo highly sensitive EMT-6 cells (Figure 1E). Taken together, these data suggest that GTN improved doxorubicin-mediated tumor regression, likely not via a direct action on cancer cells.

### 3.2. CD8 TILs Are Essential for the Anti-Tumor Efficiency of the Doxorubicin/GTN Combination

To investigate whether GTN improved the anti-tumor efficacy of doxorubicin via activation of the immune system, and more particularly CD8 TILs, we tested the impact of an anti-CD8 antibody (ab) on the anti-tumor activity of the treatment combination. The doxorubicin/GTN treatment failed to reduce tumor growth when associated with an anti-CD8 ab in the 4T1 TNBC model (Figure 2A). Accordingly, the inhibition of tumor growth induced by the doxorubicin/GTN combination was associated with an increase in the quantity of intratumor CD8 TILs (Figure 2B,C). As the expression of PD-1 on CD8 TILs represents one of the hallmarks of their activation, we analyzed PD-1 expression on CD8^+^ T Lymphocytes both in vivo and in vitro. We showed that GTN but not doxorubicin increases the expression of PD-1 on intratumor and naive CD8^+^ T lymphocytes cultured in vitro (Figure 2D,E). The expression of PD-1 on these cells may be the cause of their exhaustion in response to their engagement by PD-L1 expressed, among others, by tumor cells. However, treatment with doxorubicin and/or GTN induced a decrease in PD-L1 expression in tumor cells in vivo (Figure 2F), but also in vitro (Figure 2G; Appendix A). We assessed the importance of the PD-1/PD-L1 axis in the anti-tumor efficacy of our combination treatment by adding a PD-1 blocking antibody in our TNBC model. We observed that even if the addition of anti-PD-1 ab to doxorubicin alone or with GTN slowed tumor growth, this effect was not significant and less than that observed with the doxorubicin/GTN combination alone (Figure 2H). Thus, associating doxorubicin with GTN therefore induces an anti-tumor effect as or even more effective than its association with an anti-PD-1 ab, with this effect being dependent on CD8 TILs.

### 3.3. Doxorubicin/GTN Combination Increases the Intratumor Level of PMN-MDSCs with a Low Immunosuppressive Activity

The overactivation of CD8 TILs in the presence of GTN, confirmed by their expression of PD-1, could occur via a direct action of GTN on CD8 TILs or indirectly through the inhibition of immunosuppressive cells. Indeed, a recent study has shown that the therapeutic effect of anti-PD-1 is also highly dependent on the expression of PD-L1 on MDSCs, a key mechanism involved in their immunosuppressive activity [24]. To verify this hypothesis, we analyzed the intratumor infiltration of MDSCs. We showed no significant modulation of the level of total MDSCs in response to treatments alone or combined (Figure 3A). However, the analysis of the phenotype of these MDSCs showed that doxorubicin alone or in combination with GTN induced an increase in the intratumor PMN-MDSC/M-MDSC ratio after the second (D16) and the third (D23) injection of doxorubicin (Figure 3B), confirmed by a decrease in M-MDSC but an increase in PMN-MDSC tumor infiltration (Appendix A). In addition, doxorubicin and/or GTN greatly reduced the level of PD-L1 expression on intratumor MDSCs. This modification affected PMN-MDSCs more particularly, but not M-MDSCs (Appendix A). This decrease in the level of PD-L1 in response to doxorubicin associated or not with GTN was also found in a murine MDSC cell line, MSC2 (Appendix A). To show how the drugs affected MDSCs, we developed an RNAseq analysis on treated and untreated tumors. RNAseq analysis of 4T1 mouse tumors (TNBC) also showed that treatment with doxorubicin/GTN combination significantly decreases the expression of genes associated with MDSCs such as *s100a9* (a member of the S100 family of inflammatory mediators that serves as an autocrine feedback loop that sustains accumulation and activity of MDSC) [25] and cxcr2, a chemokine receptor involved in recruitment of MDSCs to the tumor [26] (Figure 3C; Appendix A). Indeed, we showed that the GTN/doxorubicin combination tendentially dampened *nos2* expression, involved in the immunosuppressive function of MDSC [27] (Figure 3C). A single-cell RNAseq analysis of MDSC would have been necessary to show a significant decrease. However, reduced expression of *cxcr2* and *nos2* genes by GTN and to a lesser extent by doxorubicin was confirmed via RT-qPCR on the MSC2 cell line (Figure 3D). We then addressed whether these latter GTN/doxorubicin-mediated modifications on MDSCs may affect their immunosuppressive function. We showed that doxorubicin/GTN treatment of tumor PMN-MDSCs partially alleviates their immunosuppressive activity on CD8 TILs, evidenced by the increase in CD8 TILs proliferation when co-cultured with doxorubicin/GTN-treated-PMN-MDSCs, compared to their proliferation when co-cultured with control PMN-MDSCs (Figure 3E). The enhancement of doxorubicin anti-tumor activity by GTN is therefore correlated with an increase in the level of intratumor PMN-MDSCs with a weak immunosuppressive potential.

### 3.4. The Doxorubicin/GTN Combination Regulates the FATP2 Signaling Pathway in PMN-MDSCs

We further studied the impact of doxorubicin and GTN on the immunosuppressive functions of PMN-MDSCs by analyzing the expression of FATP2, another protein whose expression is positively correlated with PMN-MDSC immunosuppressive potential [28]. We showed via RT-qPCR in the MSC2 cell line that only the doxorubicin/GTN combination significantly decreased the fatp2 expression (Figure 4A). Further, GTN and GTN +/− doxorubicin but not doxorubicin alone affected the total protein level of FATP2 on MSC2 cells (Figure 4B,C). The decrease in cell membrane FATP2 level was then confirmed in PMN-MDSCs isolated from 4T1 tumors after in vitro treatment with GTN or GTN +/− doxorubicin (Figure 4D).

Since the expression of FATP2 in PMN-MDSCs involves the activation of the transcription factor STAT5 [28], we studied whether doxorubicin and GTN affected its expression. We showed that in MSC2 cells, the GTN/doxorubicin combination dampened the phosphorylated STAT5 (p-STAT5) level (Figure 5A). We also observed in MSC2 cells that GTN but not doxorubicin induced a cleavage of the STAT5 protein, as confirmed by the appearance of a band of about 50 kDa (Figure 5B). This cleavage was detected using two different antibodies excluding the risk of a non-specific interaction (Appendix A). Both the decrease in the level and the cleavage of STAT5 appeared specific for the MDSC cell line since GTN +/− doxorubicin did not induce them in the CD8^+^ T lymphocytes from naive mice (Appendix A).

It is known that doxorubicin increases the ROS level in several types of cancer cells [29]. It is also documented that NO can interact with ROS to form radical nitrated species (RNS) responsible for post-translational modifications such as tyrosine nitration or cysteine S-nitrosylation on many proteins [30]. We then tested whether doxorubicin and/or GTN affected ROS production in MSC2 cells and whether they influenced STAT5 expression. We showed that doxorubicin and/or GTN induce the production of ROS in these cells (Appendix A). In addition, the generated ROS seem to be responsible for the STAT5 cleavage since the ROS scavenger, NAC, inhibited GTN and GTN/doxorubicin-mediated STAT5 cleavage (Figure 5C). We previously reported that GTN S-nitrosylated some proteins such as death receptors [16]. We then tested whether GTN S-nitrosylated STAT5. Using the biotin switch assay technique, we showed that GTN and another NO donor, CysNO, increased the S-nitrosylation of STAT5 (SNO-STAT5 and SNO-cleaved STAT5) in MSC2 cells, which was also dampened by NAC (Figure 5D). Moreover, we observed that GTN and CysNO-mediated cleaved form of STAT5 was also S-nitrosylated and that GTN-mediated STAT5 S-nitrosylation and cleavage was inhibited by NAC (Figure 5D). Interestingly, NAC overcame GTN and GTN/doxorubicin-mediated FATP2 protein expression both in MSC2 cells and in PMN-MDSCs from 4T1 tumors (Figure 5E,F). Finally, we tested the impact of NAC on the anti-tumor effect of the doxorubicin/GTN combination. As illustrated in Figure 5G, the NAC treatment of 4T1 tumor-bearing mice significantly alleviated doxorubicin/GTN-mediated tumor growth inhibition. Altogether, these results indicate that GTN and GTN/doxorubicin affect the ROS-mediated cleavage of STAT5, likely lead to the regulation of FATP2 and underlie the increased efficacy of doxorubicin/GTN against tumor growth.

### 3.5. PMN Gene Signature in Tumors Is Associated with Positive Clinical Outcome in TNBC Patients

It emerges from our study that doxorubicin and GTN combination treatment induces a less immunosuppressive tumor microenvironment that seems to promote an anti-tumor immune response. Transposed to humans, this immune landscape could have a beneficial impact on the prognosis for TNBC. Indeed, the modifications induced by the doxorubicin/GTN combination on the tumor microenvironment and on the PMN-MDSCs (FATP2, CXCR2, NOS2, S100A9) are in favor of reprogramming these cells towards a less immunosuppressive PMN type phenotype [31]. Consequently, we performed a RNAseq analysis on TNBC tumors obtained via needle core biopsy before and after one course of chemotherapy from 26 patients to determine whether the modulation of PMNs/PMN-MDSCs activity and tumors recruitment could be correlated with a positive clinical outcome (Figure 6; Appendix A). The expression of 32 genes related to PMNs has been analyzed before and after one course of epirubicin-based treatment in a cohort of 26 patients with TNBC composed of good (n = 8) and non- (n = 18) responders to this combination treatment. Some genes were upregulated or downregulated before or after one course (Appendix A); however, only five genes were upregulated specifically in good responders (FCAR, HPSE, TECPR2, TNFRSF10C and CD33). This signature is associated with PMNs-specific activities including neutrophil degranulation, neutrophil activation involved in immune response and neutrophil-mediated immunity. Moreover, two genes were upregulated in non-responder (HIST1H2BC and PDE4B) that are associated with the response to lipopolysaccharide (Figure 6; Appendix A). Thus, patients who respond well to chemotherapies present a molecular signature specific to the activation of neutrophils.

## 4. Discussion

Chemotherapy is the standard TNBC patients’ treatment, which can improve cure rates. However, more than a third of patients still relapse [3]. This observation laid the foundation for the rationale of combining chemotherapy with immunotherapy. Several randomized trials combining chemotherapy with immune checkpoint inhibitors (ICI) have been subsequently initiated in TNBC, particularly in early TNBC [32,33]. Despite the benefit of ICI, some patients remain resistant to these combinations, requiring the development of new combinations.

The results of this study highlight the ability of the NO donor, GTN, to potentiate the anti-tumor efficacy of chemotherapy, such as doxorubicin, in two models of BC including a TNBC subtype (4T1 cells). This potentiating effect of NO donors on the anti-tumor efficacy of chemotherapy has already been demonstrated in vitro but also in pre-clinical models of solid cancers [34,35]. However, in these different studies, the demonstration of the efficacy of NO donors was independent on their immunomodulatory effect since the tumor cells were xenografted in nude mice. To date, only one study has shown that a NO donor as GTN could potentiate the anti-tumor effect of pemetrexed in a syngraft model of non-small cell lung cancer (LLC1 cells) transplanted into C57/BL6 mice [36]. The anti-tumor effect of their therapeutic combination was then dependent on the increase in cancer cell death. However, in our study, no synergistic effect on the cell death could be demonstrated by the doxorubicin/GTN combination in vitro in the two models of BC. The enhancing effect of GTN on the doxorubicin anti-tumor efficacy in TNBC tumors seems to be dependent on the immune system, with this effect being lost in the presence of an ab blocking CD8+ T lymphocytes. In addition, flow cytometry and immuno-histo fluorescence analysis revealed that the doxorubicin/GTN combination increases TNBC infiltration by CD8 TILs. This ability to recruit CD8 TILs into the tumor represents an important feature since many studies highlight the correlation between this infiltrate and the response to treatments. Therefore, a meta-analysis, realized on approximately 3000 HER2+ or TNBC patients, showed that a high level of these cells in the tumors represented a biomarker for the prognosis of survival of these patients and would be predictive of anthracyclines’ response [37]. In addition, the analysis of several clinical trials highlighted the role of tumor CD8 TILs infiltration in anthracyclines’ therapeutic efficacy [38,39]. Indeed, these cells can recognize and kill tumor targets as demonstrated in melanoma [40,41]. The level of PD-1 expression on these CD8 TILs also represents an important factor. In fact, in chronic infections but also in cancer, PD-1 expression on CD8 TILs has been considered as a surrogate marker of T cell exhaustion [42]. However, in our study, the anti-tumor effect of GTN/doxorubicin combination correlated with the increase in this CD8 PD-1+ TILs infiltrate. Indeed, the expression of PD-1 in CD8 TILs has also been described as a CD8^+^ T cell activation marker. Studies carried out using human TNBC tumors have thus shown that these CD8 PD-1+ TILs can still produce effector cytokines and can degranulate [42]. Moreover, the infiltration of these CD8 PD-1+ TILs in breast tumors has been shown to be a factor of neoadjuvant chemotherapies +/− anti-PD-1 effectiveness [43]. In our study, the combination of an anti-PD-1 with doxorubicin only slightly enhanced the effect of this chemotherapy while the combination of doxorubicin with GTN induced a significant increase in tumor efficacy. Another biomarker of anti-PD-1 therapies efficacy is based on the expression of PD-L1 on tumor cells, which is induced in response to CD8 TILs activation and IFNγ production [39]. Indeed, in our study, the lack of efficacy of the anti-PD-1 combination with doxorubicin alone or associated with GTN is correlated with a strong decrease in the level of PD-L1 on tumor cells in vivo and in vitro. In addition, a recent study has shown that the therapeutic effect of anti-PD-1 is also highly dependent on the expression of PD-L1 on host cells and mainly on MDSCs [24]. However, the expression of PD-L1 on MDSCs and more particularly on PMN-MDSCs is inhibited in the tumors of mice treated with the doxorubicin/GTN combination. The decrease in PD-L1 levels on both tumor cells and PMN-MDSCs could therefore explain the lack of efficacy of anti-PD-1 associated with doxorubicin/GTN.

MDSCs are generated in the bone marrow and their recruitment to the tumor site is dependent on diverse set of chemokines, such as CCL2, CCL3 and CCL4, for the recruitment of M-MDSCs via the C-C chemokine receptor 2 (CCR2) [44] and the ligands of the CXC-chemokine receptor 2 (CXCR2) for PMN-MDSC [45]. Surprisingly, while the expression of cxcr2 is decreased in the tumors of mice treated with the treatment combination, the level of PMN-MDSCs is increased. Interestingly, these cells are a population of pathologically activated PMN neutrophils that are critically important for the regulation of cancer immune responses. Their presence usually correlates with poor prognostic and therapeutic outcomes [28,46] and the appearance of metastasis [47]. The main factors associated with PMN-MDSCs tumor infiltration and especially with their immune suppression activity include S100A8/9 and iNOS expression [48,49,50]. In our study, the anti-tumor effect of the doxorubicin/GTN combination was correlated with a decrease in the expression of s100a9 (in tumors) and iNOS (in MDSCs), and with a decrease in PMN-MDSCs immunosuppressive function. Moreover, the intra-tumor increase in PMN-MDSC is not linked to an increase in immunosuppression as these cells do not exhibit their typical immunosuppressive functions. Recently, Gabrilovich’s team demonstrated that the upregulation of FATP2 expression was able to reprogram PMNs [28]. Indeed, they reported that highly immunosuppressive PMN-MDSCs up-regulate FATP2, and this over-expression was dependent on the transcription factor STAT5′s activation. Moreover, the deletion of FATP2 or its selective pharmacological inhibition (lipofermata) abrogated the immunosuppressive activity of PMN-MDSCs and substantially delayed tumor progression. Accordingly, we showed that the overexpression of FATP2 is inhibited in MSC2 cells but also in tumor-infiltrated PMN-MDSCs in response to GTN. This decrease in FATP2 level is associated with the inhibition of STAT5 phosphorylation and the cleavage of STAT5 releasing a protein of about 50 kDa in MSC2 cells. STAT5 cleavage has previously been reported in prostate cancer cells. This short form of the transcription factor possessed a constitutive transcriptional activity [51]. This STAT5 cleavage has also been shown in myeloid progenitors and was highlighted to play a major role in their differentiation [52]. Several cleavage mechanisms have been proposed with the involvement of calpains [53] or proteasome [54]. In our MDSC cell model, this STAT5 cleavage appears to be independent of the action of calpains and proteasome as demonstrated by the use of ALLN, a cell-permeable inhibitor of calpain I, calpain II, cathepsin B, cathepsin L and proteasome. This cleavage was not dependent on caspases either as shown using of a caspase inhibitor VI, the Z-VAD-FMK. With the immunosuppressive function of PMN-MDSCs being largely dependent on the presence of ROS [55], the cleavage of STAT5 was tested in the presence of an antioxidant such as the N-acetyl cysteine (NAC). Interestingly, the STAT5 cleavage induced with GTN alone or associated with doxorubicin is totally inhibited by the presence of NAC. This STAT5 cleavage inhibition is concurrent with a decrease in both the membrane level of FATP2 (MSC2 cells and PMN-MDSC) in vitro and the anti-tumor effect of doxorubicin/GTN combination in vivo. In fact, the treatment of MDSCs with doxorubicin or GTN alone, but also with their combination, induces a strong production of ROS in these cells as shown via DCFH2DA detection analysis. Several studies highlighted the involvement of ROS in the appearance of NO-dependent post-translational modifications, such as S-nitrosylation. GTN induces the S-nitrosylation of STAT5 and the appearance of the cleaved form of S-nitrosylated STAT5. The involvement of ROS in this S-nitrosylation mechanism has also been proven by the inhibition of this mechanism in the presence of NAC. The relationship between STAT5, NO and ROS seems very close since Joseph et al. recently showed that STAT5 interacts with the iNOS promoter to induce its expression and thus the production of NO [56]. Thus, the inhibition of STAT5 affects the expression of iNOS and increases ROS, as also demonstrated by treating MSC2 cells by the doxorubicin/GTN combination.

Thus, the improvement in the anti-tumor efficacy of doxorubicin induced by the association with GTN is due to the ability of the GTN/doxorubicin combination to increase the intratumor recruitment of CD8 PD-1+ TILs and of PMN-MDSCs PD-L1^low^. This combination of treatment also has the ability to reprogram, in the presence of ROS, these PMN-MDSCs towards a less immunosuppressive phenotype via the STAT5 cleavage and the decrease in FATP2 expression.

This study allowed us to demonstrate the interest of associating a NO donor such as GTN with chemotherapy to enhance its anti-tumor activity. This treatment’s combination induces an increase in both tumor-infiltration and activation of CD8 TILs. These modifications seem to be correlated with PMN-MDSCs reprogramming. Indeed, PMN-MDSCs had their immunosuppressive activity lost through the ROS-dependent STAT5 inhibition. Targeting STAT5 in the treatment of cancer such as osteosarcoma has already been proposed with the use of inhibitors such as pimozide [57]. However, the use of these inhibitors must be validated in humans. The advantage of that therapeutic combination is that doxorubicin is a chemotherapy used in the treatment of TNBC and that GTN is a drug already used in the treatment of angina pectoris in Human. Moreover, several clinical trials have demonstrated the beneficial effects of the use of GTN in combination with standard chemotherapy, associated or not with radiotherapy, in patients with a non-small cell lung cancer or prostate cancer [11,12,13]. However, this is the first time that GTN’s ability to modulate the immunosuppressive activity of PMN-MDSCs is described. The modulation of the pro-tumor activity of these cells represents a target of choice as demonstrated by Veglia et al., with the reprogramming of these cells into neutrophils by inhibiting the expression of FATP2 [28]. Indeed, the activity of neutrophils seems to be a factor correlated with the response to chemotherapy in patients with TNBC. The good responders presented a PMN with neutrophil degranulation, neutrophil activation involved in immune response and neutrophil-mediated immunity.

## 5. Conclusions

In this study, we have demonstrated the importance of combining GTN with doxorubicin to increase the therapeutic efficacy of this chemotherapy in breast cancer. The lack of toxicity of GTN in combination with chemotherapies was demonstrated in a Phase II clinical trial. These findings could open a new therapeutic perspective by combining GTN to doxorubicin for patients with TNBC.

## Figures and Tables

**Figure 1 cancers-15-03129-f001:**
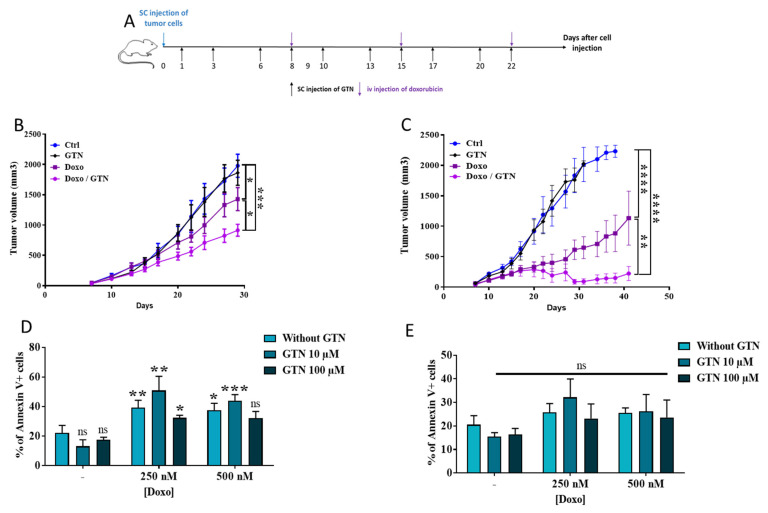
Anti-tumor effect of doxorubicin/GTN combination in two breast cancer models. (**A**) Planning of in vivo experiments. Balb/C mice bearing subcutaneous mammary tumors (5 × 10^5^ 4T1 or EMT6 cells) were treated with GTN (5 mg/kg, s.c.) the day after cancer cell injection (renewed three times per week), followed by doxorubicin (Doxo, 8 mg/kg, i.v.) when the tumor volume reached approximately 50 mm^3^ (renewed every 7 days). Tumor growth was monitored 3 times per week. (**B**,**C**). Study of the anti-tumor effect of doxorubicin, GTN as well as their combination in 4T1 ((**B**), n = 10) or EMT6 ((**C**), n = 5) mammary tumor-bearing mice treated with doxorubicin +/− GTN according to the injection scheme presented in A. Mice were sacrificed once tumors reached a volume of around 2000 mm^3^. (**D**,**E**) Five hundred thousand 4T1 (**D**) or EMT6 (**E**) mammary cancer cells were treated with doxorubicin (250 or 500 nM) +/− GTN (10 or 100 µM) for 24 h. The cell death induced by these molecules was evaluated via annexin V/7AAD labeling and analyzed using flow cytometry (n = 4). Statistical analyses were performed via a two-way ANOVA (**B**,**C**) or *t*-test (**D**,**E**): ns: not significant, * *p* ≤ 0.05, ** *p* ≤ 0.01, *** *p* ≤ 0.001, **** *p* ≤ 0.0001. The statistics shown in (**D**,**E**) represent significant differences relative to Ctrl.

**Figure 2 cancers-15-03129-f002:**
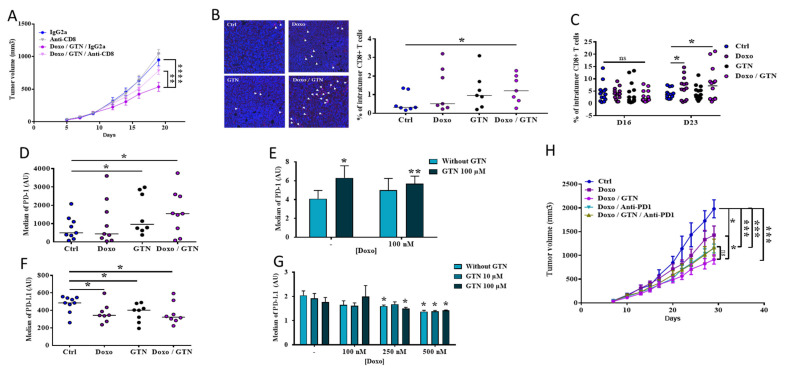
Involvement of CD8^+^ cells and PD-1/PD-L1 in the anti-tumor effect of doxorubicin/GTN combination. (**A**) Study of the anti-tumor effect of doxorubicin/GTN in combination or not with anti-CD8 in mice bearing 4T1 mammary cells (according to the injection scheme presented in Figure 1A). The anti-CD8 ab or IgG2a control isotype (Ctrl) injection started the day before the first doxorubicin (Doxo) injection (once a week, 5 mg/mL, i.p.) (n = 10). (**B**,**C**). IHC ((**B**), n = 8 mice) and flow cytometry ((**C**), n = 14 mice) analysis of the intra-tumor infiltration of CD8 TILs in response to doxorubicin +/− GTN on D16 (**C**) and/or D23 (**B**,**C**) post-injection of 4T1 tumor cells into Balb/C mice (see Figure 1A). (**D**) Flow cytometry analysis of PD-1 expression on CD8^+^-infiltrating tumors on D16, in response to doxorubicin +/− GTN according to the injection scheme presented in Figure 1A (n = 9). E. Flow cytometry analysis of PD-1 expression on CD8+ cells, recovered from spleens and lymph nodes of naive mice, and treated in vitro with doxorubicin (100 or 250nM) and/or GTN (100 μM) for 24 h (n = 3). (**F**,**G**). Flow cytometry analysis of PD-L1 expression in 4T1 tumors recovered on D16 from Balb/C mice treated with doxorubicin +/− GTN according to the injection scheme presented in Figure 1A ((**F**), n = 9), or on 4T1 cells treated in vitro with doxorubicin (100, 250 or 500nM) +/− GTN (10 or 100 μM) for 24 h ((**G**), n = 4). The bars (**D**,**F**) and histograms (**E**,**G**) represent the average of median fluorescence intensity (MFI) values. (**H**) Anti-tumor effect of doxorubicin +/− GTN in combination or not with anti-PD-1, in 4T1 mammary tumor-bearing mice (according to the injection scheme presented in Figure 1A). The anti-PD-1 treatment started the day after the first injection of doxorubicin (twice a week, 10 mg/kg, i.p.) (n = 10). Statistical analyses via two-way ANOVA (**A**,**H**) or *t* test (**B**–**G**): ns: not significant, * *p* ≤ 0.05, ** *p* ≤ 0.01, *** *p* ≤ 0.001, **** *p* ≤ 0.0001. The statistics shown in (**E**–**G**) represent significant differences relative to Ctrl.

**Figure 3 cancers-15-03129-f003:**
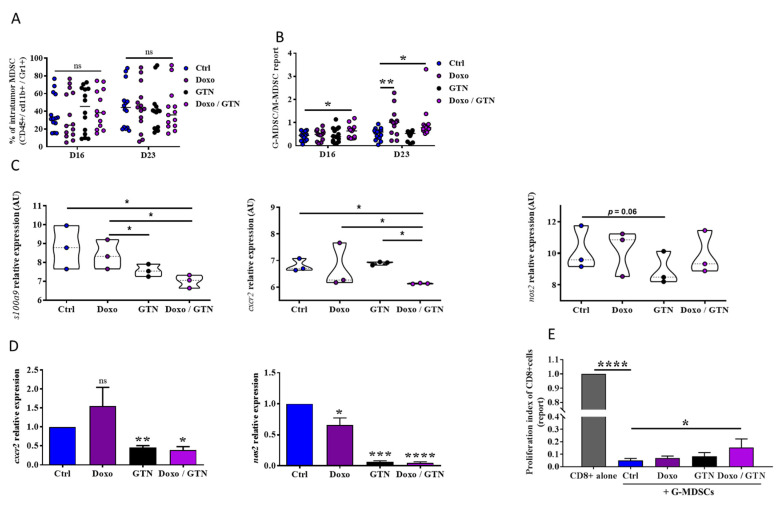
Modulation of MDSC phenotype and activity in response to doxorubicin/GTN combination. A-B. Flow cytometry analysis (n = 14 mice) of the intra-tumor infiltration by MDSC (**A**) and the G-MDSC/M-MDSC ratio (**B**) in response to doxorubicin +/− GTN on D16 and D23 post-injection of 4T1 cells into Balb/C mice (see Figure 1A). (**C**) RNAseq analysis of the intra-tumor expression of genes related to PMN/MDSC in response to doxorubicin +/− GTN on D16 post-injection of 4T1 cells into Balb/C mice (n = 3 mice) (see Figure 1A). (**D**) RT-qPCR analysis of *cxcr2* and *nos2* genes expression in MSC2 cells treated for 24 h with doxorubicin (100 nM) +/− GTN (100 µM) (n = 3). (**E**) Flow cytometry analysis of the immunosuppression induced by G-MDSC on anti-CD3/CD28 beads activated CD8+ cells after 5 days of co-culture at 2:1 ratio (n = 6 mice for the Ctrl group, and n = 3 mice for the treated groups). G-MDSC and CD8^+^ cells were, respectively, purified on D23 from 4T1 tumors of mice treated or not (Ctrl) with doxorubicin +/− GTN (see Figure 1A) and from naive mice spleens and lymph nodes. Statistical analyses via *t* test: * *p* ≤ 0.05, ** *p* ≤ 0.01, *** *p* ≤ 0.001, **** *p* ≤ 0.0001. The statistics shown in (**D**) represent significant differences relative to Ctrl.

**Figure 4 cancers-15-03129-f004:**
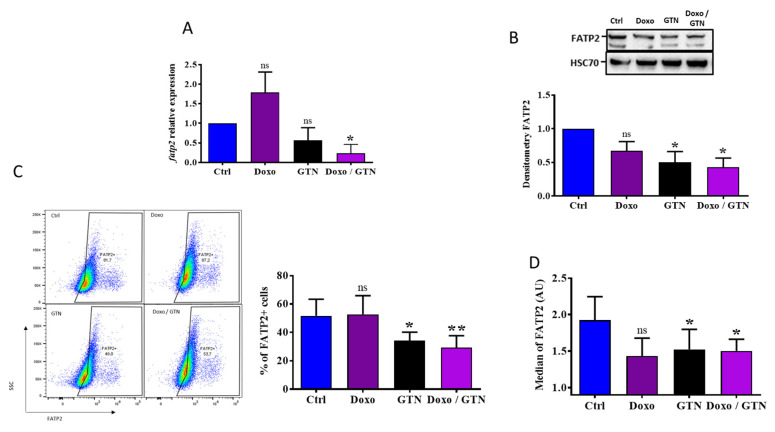
Doxorubicin/GTN combination induces a decrease in the expression of FATP2 in G-MDSCs. (**A**,**B**). Analysis of FATP2 gene ((**A**), RT-qPCR, n = 3) and protein ((**B**), Western-blot, n = 3) expression in MSC2 cells treated for 24 h (**A**) or 48 h (**B**) with doxorubicin (100 nM) +/− GTN (100 µM). Detection of anti-Hsc70 antibody served as a load control for Western blot experiments. Densitometry is shown below the blot(**B**). (**C**,**D**). Flow cytometry analysis of FATP2 membrane expression in MSC2 cells ((**C**), n = 4) or G-MDSCs purified from untreated 4T1 tumor-bearing mice ((**D**), n = 5 mice) and treated in vitro with doxorubicin (100 nM) +/− GTN (100 µM) for 24 h. The histograms represent the average of median fluorescence intensity (MFI) values (**D**). Statistical analysis via *t* test: * *p* ≤ 0.05, ** *p* ≤ 0.01. The statistics shown in all figures represent significant differences relative to Ctrl.

**Figure 5 cancers-15-03129-f005:**
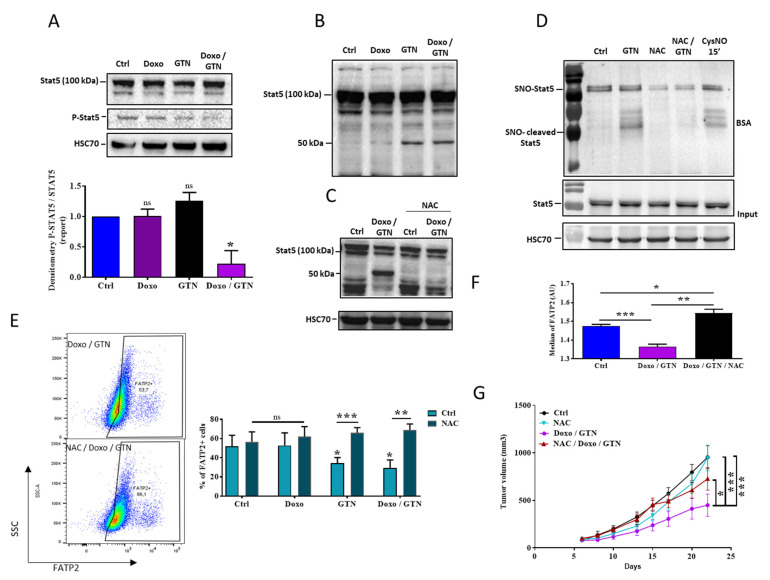
Modulation of the STAT5 pathway in response to GTN associated or not with doxorubicin. (**A**–**C**) Western blot analysis of STAT5/p-STAT5 ((**A**), n = 3) and cleaved STAT5 in MSC2 (**B**,**C**) treated with doxorubicin (100 nM) +/− GTN (100 μM) for 48 h, with or without NAC (10 mM, (**C**)). Detection of HSC70 serves as a load control for all Western blot experiments (**A**–**C**). A densitometry of the blot is shown below the blot (**A**). (**D**) Western blot and biotin switch assay analysis of STAT5 S-nitrosylation in MSC2 treated with GTN (100 μM) for 24 h, with or without NAC (10 mM, D) (**E**,**F**) Flow cytometry analysis of FATP2 membrane expression in MSC2 cells ((**E**), n = 4) or G-MDSCs purified from untreated 4T1 tumors bearing mice ((**F**), n = 5 mice) treated in vitro for 24 h with doxorubicin (100 nM) +/− GTN (100 µM) +/− NAC (10 mM). (**G**) Anti-tumor effect of doxorubicin/GTN combination in association or not with NAC, in 4T1 mammary tumor-bearing mice treated with doxorubicin/GTN (according to the injection scheme presented in Figure 1A) +/− NAC. The NAC treatment started the day before tumor cell injection (three times a week, 3.5 mg/mL, in drinking water) (n = 7 mice). Statistical analyses using *t* test (**A**,**E**,**F**) or two-way ANOVA (**G**): ns: not significant, * *p* ≤ 0.05, ** *p* ≤ 0.01, *** *p* ≤ 0.001. The statistics shown in (**A**) represent significant differences relative to Ctrl.

**Figure 6 cancers-15-03129-f006:**
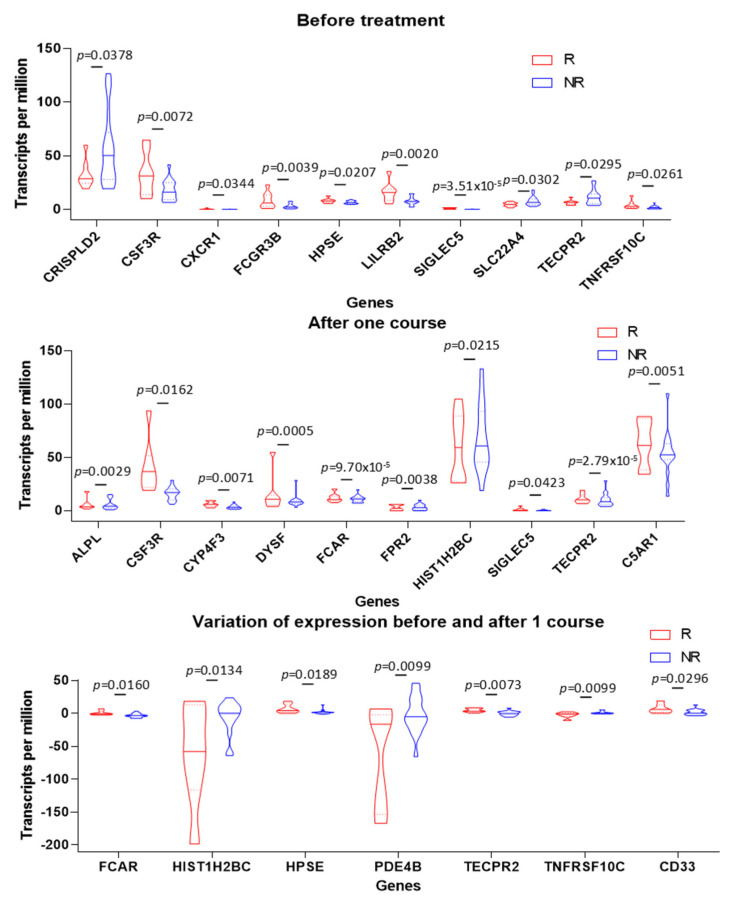
Variation of gene expression before and after the first dose of FEC/docetaxel chemotherapy. RNAseq analysis of the intra-tumor expression of genes related to PMN cells, in good (red) and non-responder (blue) TNBC patients, before and after the first course of 5-Fluorouracil/Epirubicin/Cyclophosphamide/Docetaxel chemotherapy. Statistical analyses were performed by *t* test.

## Data Availability

The datasets used and/or analyzed during the current study are available from the corresponding author on reasonable request.

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
