# Peer review of "GTN Enhances Antitumor Effects of Doxorubicin in TNBC by Targeting the Immunosuppressive Activity of PMN-MDSC"

_cancers, 2023, doi:10.3390/cancers15123129_

Round 1

Reviewer 1 Report

Comments to authors for their considerations: 

Abstract:

The abstract might be strengthened by including more information about the significance of the findings. The immunosuppressive tumor microenvironment is a key hurdle to effective treatment, and the relevance of TNBC as an aggressive and difficult-to-treat form of breast cancer might be briefly mentioned in the abstract.

It would be beneficial to include information about the sample size and the amounts of GTN and doxorubicin used if the abstract is to be improved further.

Introduction:

More information about the restrictions and difficulties of using GTN in conjunction with chemotherapy would be helpful in this section. Increased toxicity and harmful consequences are two examples that need to be addressed.

Possible practical applications of these findings might be discussed, such as the feasibility of conducting future clinical trials examining the combination of GTN and doxorubicin in TNBC patients.

Materials and Methods

There is some information about the animal models utilized, such as the ages and sexes of the mice, but more details, such as the strain and origin of the mice, would be helpful.

More particular information on the methods utilized for isolating and labeling T cells and PMN-MDSCs from primary mouse immune cells would be helpful.

For the RT-qPCR analysis, it would be beneficial to provide more information on the precise primer sequences utilized and to indicate which genes or markers were being studied.

Results

No introduction or explanation of the study issue or hypothesis being tested is provided. It would be beneficial to identify the study's overarching goal and explain how the experiments were designed to address this question.

It would be helpful if the results were described in more precise and detailed language. Comparing the 4T1 TNBC model to the estrogen and progestin receptor-positive EMT-6 cell line, for instance, the rate at which tumor development is inhibited is lower in the 4T1 TNBC model.

There is a need for improvement in the figures and legends to better explain the methodology and findings of the experiments. It would be beneficial, for instance, to provide a key in the caption explaining the various treatment groups depicted in Figure 1A.

What are the next steps for additional research into the use of GTN in combination with doxorubicin for the treatment of breast cancer, and what are the possible therapeutic applications of doing so?

Discussion

This section may be strengthened by including more information about the structure and methodology of the study. Results from experiments are described; however, neither the methodology nor any statistical analyses used to arrive at those results are provided. Readers will be better able to assess the study's validity and reliability if more information is provided regarding the study's design and methods, as well as statistical analysis.

The study's shortcomings may also need some additional explanation from the authors. In particular, they note that the synergistic effect of GTN on the anti-tumor efficacy of chemotherapy in vitro was not established, but they do not provide any explanation for why this would be the case or how this limitation might affect the larger implications of their findings. Readers might benefit from more information regarding the study's breadth and generalizability if it were presented.

The authors may want to consider changing some of the terminology to make the section more user-friendly. The part has a sophisticated, scientific tone, although some of the language is technical and may be challenging for non-experts to follow. The section may be made more approachable to a larger readership by streamlining some of the terminology and offering greater context and explanations for key words.

Conclusions

Authors might want to rethink some of the terminology in the final section, the conclusions, to make sure it doesn't make any assertions that can't be backed up by the data. Without additional confirmation and replication of the results in larger clinical studies, it may be too soon to declare that your findings "open up a new therapeutic perspective."

Author Response

Comments to authors for their considerations: 

Abstract:

The abstract might be strengthened by including more information about the significance of the findings. The immunosuppressive tumor microenvironment is a key hurdle to effective treatment, and the relevance of TNBC as an aggressive and difficult-to-treat form of breast cancer might be briefly mentioned in the abstract.

It would be beneficial to include information about the sample size and the amounts of GTN and doxorubicin used if the abstract is to be improved further.

Thank you very much for your comments. We have taken it into account and modified the abstract. We have also added the doses of doxorubicin and GTN used in this study.

Introduction:

More information about the restrictions and difficulties of using GTN in conjunction with chemotherapy would be helpful in this section. Increased toxicity and harmful consequences are two examples that need to be addressed.

We have added a sentence (lines 83-86) mentioning the absence of toxicity of the combination of GTN and chemotherapies in the treatment of cancers.

Possible practical applications of these findings might be discussed, such as the feasibility of conducting future clinical trials examining the combination of GTN and doxorubicin in TNBC patients.

We added in the conclusion (lines 606-609) « The lack of toxicity of GTN in combination with chemotherapies was demonstrated in a Phase II clinical trial, these findings could open a new therapeutic perspective by combining GTN to doxorubicin for patients with TNBC. »

Materials and Methods

There is some information about the animal models utilized, such as the ages and sexes of the mice, but more details, such as the strain and origin of the mice, would be helpful.

The origin as well as the strain of the mice is cited in the material and method, paragraph 2.1. "Balb/C female mice of 6-8 weeks of age were purchased from Charles River Laboratories."

More particular information on the methods utilized for isolating and labeling T cells and PMN-MDSCs from primary mouse immune cells would be helpful.

The isolation of the primary cells is described in paragraph 2.3 of the material and methods. For PMN-MDSCs, we cited the article by Martin et al (2018) in which we use Anti-Ly-6G MicroBeads UltraPure, mouse (Miltenyi) to isolate these cells. For the isolation of T lymphocytes, we indicated that "T Lymphocytes were previously isolated from spleens and lymph nodes of naive mice using the Pan T cells isolation kit II (Miltenyi)." The Abs list used for the labelling of immune cells is shown in the supplementary Table 2.

For the RT-qPCR analysis, it would be beneficial to provide more information on the precise primer sequences utilized and to indicate which genes or markers were being studied.

The specific primer and the housekeeping genes used are shown in supplementary table 1

Results

No introduction or explanation of the study issue or hypothesis being tested is provided. It would be beneficial to identify the study's overarching goal and explain how the experiments were designed to address this question.

The aim of our study was explained in the introduction (lines 87-93). In the results section we clearly state that: « In order to test whether the NO donor GTN, can improve the therapeutic efficacy of a chemotherapy conventionally used in the treatment of BC, we used the doxorubicin/GTN combination in two BC models ».

It would be helpful if the results were described in more precise and detailed language. Comparing the 4T1 TNBC model to the estrogen and progestin receptor-positive EMT-6 cell line, for instance, the rate at which tumor development is inhibited is lower in the 4T1 TNBC model.

Thank you very much for the comment, we have clarified in the text that the EMT-6 model is a less aggressive tumor model than the TNBC 4T1 model. We have added line 240: " the triple negative 4T1 model and the estrogen and progestin receptor positive cell EMT-6 model. The latter model is a less aggressive tumor model than the triple negative model. »

There is a need for improvement in the figures and legends to better explain the methodology and findings of the experiments. It would be beneficial, for instance, to provide a key in the caption explaining the various treatment groups depicted in Figure 1A.

Thank you very much for this comment. We tried to do the best we could with our 4 groups of mice by keeping the same color per group at all times. As shown in Figure 1A, GTN is in black and doxorubicin is in purple. The combination of the 2 treatments is shown in pink from figure B and in blue for the control group. This color code is kept the same for all figures.

What are the next steps for additional research into the use of GTN in combination with doxorubicin for the treatment of breast cancer, and what are the possible therapeutic applications of doing so?

Thank you very much for this comment. The next step will be to validate all these results on other breast cancer models, TNBC but also other types of BC. In order to get closer to what is found in patients, other tumor models in mice could be used; orthotopic mammary tumors or spontaneous mammary cancer models (MMTV-neu mice)

Discussion

This section may be strengthened by including more information about the structure and methodology of the study. Results from experiments are described; however, neither the methodology nor any statistical analyses used to arrive at those results are provided. Readers will be better able to assess the study's validity and reliability if more information is provided regarding the study's design and methods, as well as statistical analysis.

The methodology and statistics used could have been added in the discussion section but we preferred to describe them in the material and methods section and in the different figure legends.

The study's shortcomings may also need some additional explanation from the authors. In particular, they note that the synergistic effect of GTN on the anti-tumor efficacy of chemotherapy in vitro was not established, but they do not provide any explanation for why this would be the case or how this limitation might affect the larger implications of their findings. Readers might benefit from more information regarding the study's breadth and generalizability if it were presented.

Thank you very much for this comment. One explanation for the discrepancy between the in vitro and in vivo results is due to the involvement of the immune system in the in vivo anti-tumor effect of the combination which is absent in the in vitro assay.

The authors may want to consider changing some of the terminology to make the section more user-friendly. The part has a sophisticated, scientific tone, although some of the language is technical and may be challenging for non-experts to follow. The section may be made more approachable to a larger readership by streamlining some of the terminology and offering greater context and explanations for key words.

Thank you very much for this proposal, immuno-oncology is always complicated to explain simply. We have tried to explain as much as possible all the technical terms necessary to understand the article.

Conclusions

Authors might want to rethink some of the terminology in the final section, the conclusions, to make sure it doesn't make any assertions that can't be backed up by the data. Without additional confirmation and replication of the results in larger clinical studies, it may be too soon to declare that your findings "open up a new therapeutic perspective."

You are absolutely right, we have modulated the discussion by stating: « These findings could open a new therapeutic perspective by combining GTN to doxorubicin for patients with TNBC. »

Reviewer 2 Report

GTN enhances antitumor effects of doxorubicin in TNBC by 2 targeting the immunosuppressive activity of PMN-MDSC

The study investigates the effectiveness of the combination therapy of doxorubicin/GTN in triple negative breast cancer. The findings will help potential readers to understand the mechanisms underlying the mode of action of the immune-mediated anticancer therapy of the combination drug studied and the possible therapeutic application of GTN- doxorubicin combination for TNBC patients. The approach and the overall design of the study are good. However, the authors should address the following concerns.

Ø  How did the authors fix the dose of GTN for mice experiments?

Ø  The proper references should be cited for the dose of each antibody used for in vivo injections.

Ø  Also, the dose of N-acetyl-L-cysteine needs to be justified.

Ø  How did the authors measure the tumor volume? Add the details.

 The materials and Methods sections need to be improved by including these comments

Ø  Add limitations of the study.

Author Response

GTN enhances antitumor effects of doxorubicin in TNBC by 2 targeting the immunosuppressive activity of PMN-MDSC

The study investigates the effectiveness of the combination therapy of doxorubicin/GTN in triple negative breast cancer. The findings will help potential readers to understand the mechanisms underlying the mode of action of the immune-mediated anticancer therapy of the combination drug studied and the possible therapeutic application of GTN- doxorubicin combination for TNBC patients. The approach and the overall design of the study are good. However, the authors should address the following concerns.

Ø  How did the authors fix the dose of GTN for mice experiments?

Thank you very much for this question, we have defined this dose in a previous article: https://doi.org/10.3390/ijms22168449. This reference is added in material and methods section [17]

Ø  The proper references should be cited for the dose of each antibody used for in vivo injections.

Thank you for pointing out this oversight. We have added the clones of anti-PD-1, anti-CD8a and IgG2a in the material and methods section.

Ø  Also, the dose of N-acetyl-L-cysteine needs to be justified.

Thank you very much for this question. We have previously used NAC in mouse cancer models in the paper doi: 10.1080/2162402X.2015.1123369.. We have added the reference in the material and methods section [18].

Ø  How did the authors measure the tumor volume? Add the details.

We have added in the text of the material and methods section: « Tumor growth monitoring was performed every 2-3 days using a caliper. The tumor volume (mm3) is determined with the formula (l2*L)/2. »

 The materials and Methods sections need to be improved by including these comments

Ø  Add limitations of the study.

The limitation of this study lies in the use of a single TNBC and a single BC model, both implanted subcutaneously. The next step will be to validate all these results on other breast cancer models, TNBC but also other types of BC. In order to get closer to what is found in patients, other tumor models in mice could be used; orthotopic mammary tumors or spontaneous mammary cancer models (MMTV-neu mice).

Reviewer 3 Report

The article by Mabrouk et al. on the enhancement of doxorubicin effect by GTN in TNBC is an interesting article. The authors deserves appreciation for conducting the thorough research in vivo. The experiments conducted by the authors are well-designed and the results are of good quality, making the paper relevant and interesting to readers. The authors also provided a proper discussion and conclusion to support their findings. I would recommend this article to be accepted for publication.

Author Response

Thank you very much for all these positive comments .